# Self-Assembled Monolayer Formation on a Dental Orthodontic Stainless Steel Wire Surface to Suppress Metal Ion Elution

**Hironori Tamaki [1], Shigeaki Abe [2],\*, Shuichi Yamagata [1], Yasuhiro Yoshida [1] and Yoshiaki Sato [1]**

1   Faculty of Dental Medicine, Hokkaido University, Kita 13, Nishi 7, Sapporo 060-8586, Japan;
    hiro.dental@me.com (H.T.); shuic@den.hokudai.ac.jp (S.Y.); yasuhiro@den.hokudai.ac.jp (Y.Y.);
    yoshion51@me.com (Y.S.)
2   Graduate School of Biomedical Sciences, Nagasaki University, Sakamoto 1-7-1, Nagasaki 852-8102, Japan
*   Correspondence: sabe_den@nagasaki-u.ac.jp

**Abstract:** Metal ion elution, including Cr and Ni from dental orthodontic stainless steel, accounts for some allergies. In this study, a self-assembled monolayer (SAM) on a wire surface is proposed for suppressing such elution. This method involves modifying the stainless steel surface using phosphonic acid containing a long alkyl chain. The uncoated and coated wires are immersed in different acidic solutions, and the supernatant is analyzed by inductively coupled plasma mass spectrometry after 1–4 weeks. The results reveal that Cr and Ni ion elution is significantly suppressed by SAM modification. These findings will help in minimizing potential allergens from dental orthodontics.

**Keywords:** surface modification; suppress metal ion elution; self-assembled monolayer; dental orthodontic stainless steel wire

## 1. Introduction

Chemical modifications, such as an organic monolayer formation, are simple and useful methods for enhancing metal and alloy surface functions [1–4]. These methods have been involved in studies on several fields, including nanoelectronics [5], optoelectronics [6], chemical and biochemical sensing [7], bioengineering [8], life sciences [9], and biomedical applications [10]. Recent developments in medical technology require new functions, such as mechanical, electrical, and optical functions and biocompatibility. Stainless steel is a material routinely used in many applications in different fields. It is a biomaterial with a bioacceptable property. Materials fabricated from it, such as prosthetic joints, medical stents and plates, and orthodontic and prosthetic products, are implanted in the body for long durations. Although stainless steel exhibits lower corrosion tolerance compared with Ti, Ti alloys, or other dental alloys, it displays excellent mechanical properties, especially in orthodontics [11]. Though it is "biotolerant", its prolonged use in a biological environment creates metal ion elution. In particular, these dental materials are exposed to varying pH, depending on metabolism in the oral environment [12].

Some metal ions in stainless steel, such as Cr, Ni, and Pb, cause metal allergies [13–16]. In the dental field, metal allergies constitute a delayed type of allergy classified as type IV [17]. These are attributed to incomplete antigens formed by the binding of Ni and Cr ions with proteins in the oral environment [18]. Therefore, selecting a stainless steel wire in orthodontic treatments sometimes requires a metal allergy check. To this point, suppressing the elution of these ions may reduce metal allergy risk.

In this study, we initially investigated chemical modifications on dental orthodontic stainless steel wire surfaces and subsequently evaluated their metal ion elution suppression. For example,

phosphoric acid and phosphonic acid, which contain alkyl chains, are known to spontaneously react with metal oxide surfaces, such as those of titanium oxide ($TiO_2$) [19], aluminum oxide ($Al_2O_3$) [20], yttria ($Y_2O_3$) [21], and stainless steel [22], forming monolayer films, which are termed self-assembled monolayer (SAM) films. We initially modified a dental stainless- steel wire surface using an SAM film, and then the modified wire was immersed in acidic solutions (hydrochloric acid [HCl], phosphoric acid [$H_3PO_4$], and lactic acid) for 1–4 weeks. In the dental field, phosphoric acid is used for edging treatment of the enamel surface, and lactic acid is generated in the oral environment by metabolism [23]. Therefore, modified dental stainless steel wires have been immersed in several acidic solutions, including those stronger than the conditions in different oral environments, for accelerated simulation of the oral environment. To estimate the metal ion elution suppression, the Cr and Ni ion concentrations in the supernatant were determined. In addition, we added a polyamine-tethered SAM on the wire using a condensation reaction. As polyamine derivatives are known to possess antibacterial properties [24,25], their usage in surface modification can decrease metal allergy risk and induce metals with antibacterial properties.

## 2. Materials and Methods

### 2.1. Materials

A dental orthodontic stainless steel wire (TRU-CHROME, 0.016″ × 0.022″) was purchased from JM Ortho Co., Ltd. (Tokyo, Japan), and octadecylphosphonic acid (ODPA) was obtained from Miyoshi Oil & Fat Co., Ltd. (Tokyo, Japan). The 10-carboxydecylphosphonic acid (HOOC-DPA) was bought from Wako Inc. (Tokyo, Japan), and *N,N″*-bis(3-aminopropyl)tetramethylenediamine phosphate hexahydrate (polyamine) was acquired from Tokyo Chemical Industry Co., Ltd. (Tokyo, Japan). The 4-(4,6-dimethoxy-1,3,5-triazin-2-yl)-4-methylmorpholinium chloride (DMT-MM), hydrochloric acid, phosphoric acid, and lactic acid were purchased from Sigma-Aldrich (St. Louis, MO, USA). The saline was purchased from Otsuka Pharmaceutical Factory (Naruto, Japan), and all chemicals were used without further purification. The chemical structure of the phosphonate derivatives and polyamine are shown in Scheme 1.

**Scheme 1.** Chemical structures of alkyl phosphates and the polyamine.

### 2.2. Surface Modification of the Stainless Steel Surface

The dental orthodontic wires were cut to 80 mm pieces and washed by sonication using a natural detergent solution, acetone, and methanol for 10 min. After rinsing using distilled water, the wires were immersed in 100 mL of 10 mM alkyl phosphonic acid/dehydrated ethanol and stirred for 2 h, followed by drying at 120 °C for 2 days. This procedure slightly modified Marder's conditions [26], as illustrated in Scheme 2.

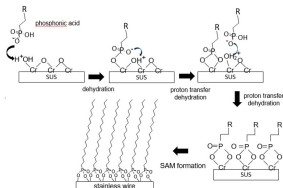

**Scheme 2.** Schematic illustration of SAM formation.

### 2.3. Polyamine Modification of the Stainless Steel Surface

To tether the polyamine on the stainless steel surface, the cleaned bare stainless steel was initially modified using HOOC-DPA. The obtained wire was subsequently modified using –COOH at the end of the SAM and by immersing it in 10 mM polyamine/dehydrated ethanol solution, accompanied by stirring for 2 h. Furthermore, 1.2 eq. of DMT-MM was added to the reaction mixture to enhance amide bond formation. This procedure was consistent with our previous studies [27], but with some minor modifications.

### 2.4. Characterization of the Obtained SAM

The surface wettability of the wire was monitored to identify the SAM formation on the stainless steel surface. First, 10 μL of distilled water was dropped on the surface at 25 °C. The contact angle was recorded using a VHD-3000 digital microscope (Keyence, Tokyo, Japan), with its lens rotated by 90°, and the sample with dropped water was observed from the horizontal direction ($n = 5$). The contact angle was then calculated from the image using the attached analysis program. The thickness of the produced SAM was estimated by a spectroscopic ellipsometer (DHA-F, Mizojiri Optics, Tokyo, Japan, wavelength = 633 nm, refractive index: $n = 2.76$, extinction coefficient: $k = 3.79$). To examine amide bond formation, a micro-Raman spectroscopic system (inVia Reflex; Renishaw K.K., Tokyo, Japan, laser excitation wavelength = 532 nm) was employed.

### 2.5. Determination of the Metal Ion Elution

To estimate metal ion elution suppression by SAM formation, the coated wires were immersed in 12 mL of 0.05 M HCl (pH = 1.4), 0.1 M $H_3PO_4$ (pH = 1.7), 1 M lactic acid (pH = 2.0), 0.1 M lactic acid (pH = 2.5), and saline (pH = 6.9) for 1–4 weeks at 37 °C while continuously stirring ($n = 5$). Subsequently, the eluted metal ions in the solutions were determined by inductively coupled plasma mass spectrometry (ICP-MS) (8800 ICP-QQQ, Agilent Technologies, Tokyo, Japan).

### 2.6. Statistical Analysis

All data are presented as the mean ± SD, with statistical differences analyzed using one-way analysis of variance (ANOVA, $p < 0.05$).

## 3. Results

### 3.1. Wettability of the Modified Stainless Steel Surface

The contact angles of the stainless steel substrate before and after SAM formation are depicted in Figure 1. Depending on the modification, the obtained contact angle was verified. When the surface

was modified using a long alkyl-chain phosphonate, the contact angle increased from $52° \pm 4°$ to $65° \pm 4°$ (Figure 1b). In contrast, the contact angle decreased when a carboxyl group was added to the end of the alkyl phosphonate (as illustrated in Figure 1c). The contact angle also increased when polyamine was tethered to the carboxyl-terminated surface, with all results presented in Table 1.

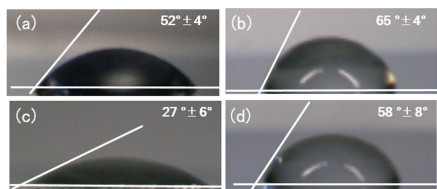

**Figure 1.** Contact angle of the water–stainless steel surface: (**a**) control, (**b**) ODPA-SAM-modified samples, (**c**) HOOC-DPA-SAM-modified samples, and (**d**) polyamine-tethered HOOC-DPA -SAM-modified samples.

**Table 1.** Contact angle of the water–stainless steel surface.

| Sample | Contact Angle |
|---|---|
| Control | $52° \pm 4°$ |
| SAM (ODPA) | $65° \pm 4°$ |
| SAM (HOOC-DPA) | $27° \pm 6°$ |
| SAM (polyamine) | $58° \pm 8°$ |

### 3.2. Suppression of Metal Ion Elution Via Surface Modification

The orthodontic dental stainless steel wire was immersed in several acidic solutions to estimate metal ion elution. After 1–4 weeks of immersion, the Cr and Ni ion concentrations in the supernatant were determined by ICP-MS. In Figure 2, the Cr and Ni ion concentrations for the $H_3PO_4$ and lactic acid solutions significantly decreased upon ODPA-SAM formation. In fact, when the bare wire was immersed in 0.1 M phosphoric acid for a week, 0.06 ppm Cr was eluted per 10 mm of wire. However, surface modification suppressed the elution by less than 0.03 ppm. For 0.1 M lactic acid, the eluted concentrations were low, with negligible difference. After four weeks of immersion, the modification changed the eluted concentration from 0.06 to 0.02 ppm. The behavior of Ni ions was consistent with that of Cr ions, with the data summarized in Table 1.

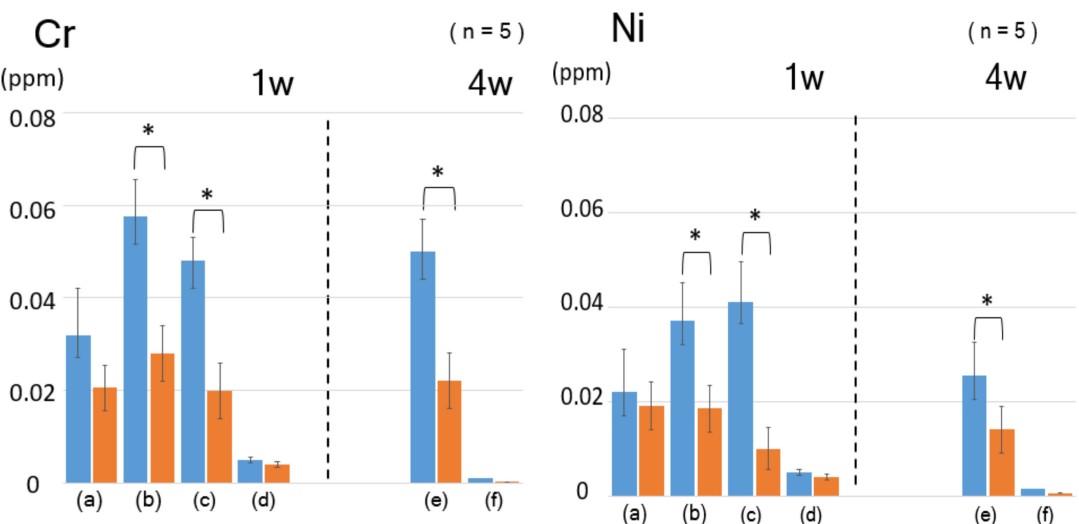

**Figure 2.** Concentration of the eluted metal ions from the ODPA-modified dental wire immersed in (**a**) 0.05 M HCl, (**b**) 0.1 M $H_3PO_4$, (**c**) 1 M lactic acid, (**d**) 0.1 M lactic acid for one week, (**e**) 0.1 M lactic acid, (**f**) saline for four weeks; left: Cr ions, right: Ni ions. ($p < 0.05$).

### 3.3. Surface Modification Using Polyamine

To provide antibacterial properties, polyamine was applied on the stainless steel surface. The surface was modified with a carboxyl group at the end of the long alkyl-chain phosphate (HOOC-DPA), as illustrated in Scheme 3, and the polyamine reacted with the carboxyl group through a condensation agent. The SAM formation was characterized by measuring the contact angle and thickness and performing Raman spectroscopy. As depicted in Figure 1, the contact angle of the SAM modified with the carboxyl group-ending long alkyl-chain phosphonate (HOOC-DPA-SAM) increased after the condensation reaction with polyamine. The thickness of the layer also increased from 1.7 ± 1.0 to 2.9 ± 0.9 nm. In addition, the Raman spectrum of the surface displayed an enhanced peak at approximately 1300 cm$^{-1}$ (assigned to the N–H groups), whereas peaks at approximately 3200 and 3400 cm$^{-1}$ decreased (assigned to the –COOH group), as depicted in Figure S1.

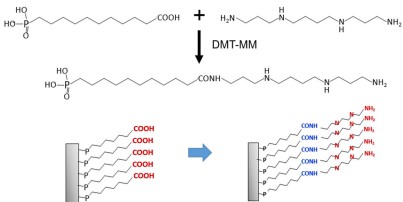

**Scheme 3.** Schematic illustration of polyamine-tethered SAM formation.

The surface-modified stainless steel wire immersed in lactic acid solution (Figure 3) demonstrated that the polyamine-modified SAM also suppressed metal ion elution. After polyamine modification, the Cr and Ni ion concentrations significantly decreased (Table 2).

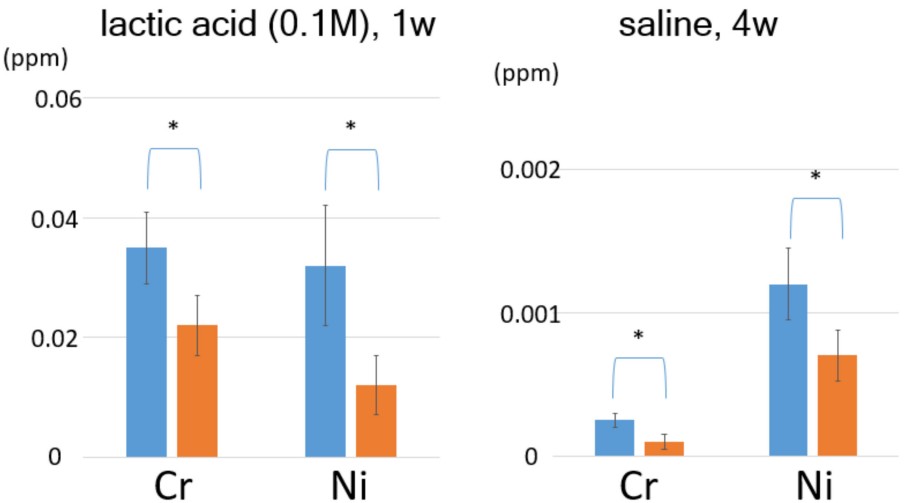

**Figure 3.** Concentration of eluted metal ions from the polyamine-modified dental wire. 1 M lactic acid for one week and 0.1 M lactic acid for four weeks; left: Cr ions, right: Ni ions. ($p < 0.05$).

**Table 2.** Concentration of eluted metal ions from the modified wire immersed to several solution.

| Immersion Period/Weeks | Concentration of Eluted Metal Ions (Mean ± SD)/ppm | | | | | |
| | [Cr] | | [Ni] | | Ratio | |
| | Control *a* | ODPA-SAM *b* | Control *c* | ODPA-SAM *d* | *b/a* | *d/c* |
|---|---|---|---|---|---|---|
| 1 | 0.032 ± 0.010 | 0.023 ± 0.006 | 0.022 ± 0.008 | 0.019 ± 0.005 | 0.719 | 0.864 |
| 1 | 0.057 ± 0.010 | 0.028 ± 0.006 | 0.037 ± 0.010 | 0.018 ± 0.005 | 0.481 | 0.486 |
| 1 | 0.046 ± 0.009 | 0.022 ± 0.006 | 0.041 ± 0.010 | 0.012 ± 0.005 | 0.477 | 0.301 |
| 1 | 0.044 ± 0.010 | 0.003 ± 0.001 | 0.005 ± 0.001 | 0.004 ± 0.001 | 0.76 | 0.778 |
| 4 | 0.025 ± 0.007 | 0.025 ± 0.007 | 0.024 ± 0.006 | 0.015 ± 0.004 | 0.588 | 0.625 |
| 4 | 0.002 ± 0.0002 | N/A | 0.001 ± 0.0002 | N/A | - | - |

## 4. Discussion

The contact angles for the stainless steel substrate before and after SAM formation are depicted in Figure 1. Generally, the mechanism of SAM formation has been proposed [22] as illustrated in Scheme 2. Depending on the modification, the obtained contact angle was verified. The contact angle during the formation of the long alkyl-chain phosphonate SAMs terminating with a carboxyl group was 27° ± 6°, increasing to 58° ± 8° after the condensation reaction with polyamine, with the layer thickness also increasing. In addition, the Raman spectra demonstrated amide bond formation and a decrease of –COOH groups (Figure S1). These results indicate that polyamine–SAM formation occurred, as shown in Scheme 3. The condensation reaction on the end of SAM occurred as described in Reference [28]. This property change has the potential of reducing dental plaque formation on the wire surface [29].

In general, polyamines are known to possess antibacterial properties [30]. Mukherjee et al. synthesized poly(hydroxymethyl methacrylates) with amino acid side chains, and the synthesized polymers displayed efficient antibacterial activity [31]. Zardinia et al. reported that surfaces of multi-walled carbon nanotubes (MWCNTs) exhibited positively-charged amino acids and produced polyamino-modified MWCNTs [32]. According to some studies, the positively-charged groups on the surfaces are responsible for adsorption, damaging the negatively-charged membranes of bacteria. Thus, the polyamine modification in this study also likely imparted antibacterial properties [33,34].

When the wires were immersed in acid solution, the ODPA modification suppressed Cr and Ni ion elution, reducing their initial concentrations by almost 50% (as shown in Table 3). For example, for the non-modified dental wire immersed in 0.1 M lactic acid for a week, the eluted Cr and Ni ion concentrations were 0.046 ± 0.009 and 0.041 ± 0.010 ppm, respectively. In contrast, the concentrations of the ions for the solution with the ODPA-modified wire were 0.022 ± 0.006 ppm for Cr and 0.012 ± 0.005 ppm for Ni, representing 47.7% and 30.1% reductions, respectively. The polyamine-modified SAM also significantly suppressed metal ion elution, such as ODPA-SAM. In the case of immersion in HCl solution, SAM formation did not inhibit elution effectively. The presence of $Cl^-$ ions are known to produce corrosion of stainless steel, called pitting corrosion (as shown in Figure S2). These corrosion are generated by Cr and Ni elution. Although no pitting corrosion was observed under the immersion conditions in this study, $Cl^-$ ions can diffuse though the SAM film because of their small size, and then can directly affect the wire's surface as an aggressive species.

**Table 3.** Concentration of eluted metal ions from the modified wire immersed to lactic acid.

| Solution | Ion | Concentration of Eluted Metal Ions (Mean ± SD)/ppm | | | Ratio | |
| --- | --- | --- | --- | --- | --- | --- |
| | | Control *a* | ODPA-SAM *b* | Polyamine-SAM *c* | *b/a* | *c/a* |
| lactic acid | [Cr] | 0.046 ± 0.009 | 0.022 ± 0.006 | 0.028 ± 0.007 | 0.477 | 0.301 |
| 1 M, 1 week | [Ni] | 0.041 ± 0.010 | 0.012 ± 0.005 | 0.023 ± 0.005 | 0.76 | 0.778 |
| lactic acid | [Cr] | 0.044 ± 0.010 | 0.025 ± 0.007 | 0.030 ± 0.006 | 0.588 | 0.682 |
| 0.1 M, 4 weeks | [Ni] | 0.024 ± 0.006 | 0.015 ± 0.004 | 0.017 ± 0.004 | 0.625 | 0.708 |

As the immersion period extended to four weeks, the eluted ion concentrations increased ca. 10 times, clearly demonstrating suppression. However, the modification effect was absent for the saline solution, even after four weeks. These results also indicate that immersion measurements in acidic conditions at saline or lower concentrations for a longer period must be envisaged.

## 5. Conclusions

In this study, we investigated two types of surface modifications on orthodontic stainless steel. The first involved coating with a long alkyl-chain monolayer, whereas the second was a polyamine monolayer. Both modifying methods significantly suppressed metal ion elution, including Cr and Ni, from the wires. This suppression had major practical significance because metals in the wires represent potential allergens. Therefore, surface modifications can be exploited for the improvement of materials in dental and other medical fields.

**Supplementary Materials:** The following are available online at http://www.mdpi.com/2079-6412/10/4/367/s1. Figure S1: Raman spectra showing the eluted metal ion polyamine-tethered reaction on the stainless steel surface, (a) before and (b) after the reaction. Figure S2: A typical SEM image of a dental orthodontic stainless steel wire surface (0.2 M HCl for one week).

**Author Contributions:** Conceptualization, H.T. and S.A.; methodology, S.A.; validation, H.T. and S.A.; investigation, H.T.; writing—original draft preparation, H.T.; writing—review and editing, S.A.; supervision, S.Y. and Y.Y.; project administration, Y.S. All authors have read and agreed to the published version of the manuscript.

**Funding:** This work was partially supported by a Grant-in-Aid for Scientific Research C (No. 18K09616) from the Ministry of Education, Culture, Sports, Science and Technology, Japan.

**Acknowledgments:** We would like to thank OPEN FACILITY (Hokkaido University Sousei Hall) for allowing us to use an inductively coupled plasma mass spectrometry for detection of eluted metal ions. We also thanks for Takayuki Kiba (Kitami Institute of Technology) for ellipsometry measurement.

**Conflicts of Interest:** The authors declare no conflict of interest.

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
