# Peer review of "Self-Assembled Monolayer Formation on a Dental Orthodontic Stainless Steel Wire Surface to Suppress Metal Ion Elution"

_coatings, doi:10.3390/coatings10040367_

Round 1

Reviewer 1 Report

The manuscript by Tamaki et al. investigates whether long alkyl chain or polyamine self-assembled monolayer (SAM) modification of stainless-steel dental wire surfaces can suppress their metal ion elution and possibly can confer anti-bacterial activity. They found that such SAM modifications could indeed significantly decrease Cr and Ni ion elution, although their anti-bacterial activity remains to be tested. The experimental design of this work is straightforward and the conclusion is generally supported by the data presented. Their findings may have implications for improving the quality of dental materials and perhaps of other biomedical devices. My comments are as follows.

1.The quality of Figure 2 and 3 can be improved by removing unnecessary labels and by adding an explanation of the asterisks on bars (P value?) and of the error bars (mean ± SD?) to the figure legends.

2.On Page 6, Line 147, it would be helpful if the authors could state precisely the percentage of reduction in elution after modifications as compared to control. The comment on "quantitative comparison" of test results applies here and elsewhere throughout the text.

Author Response

According to reviewer’s comment, manuscript is checked by native English speaker (written by red letters).

1. The quality of Figure 2 and 3 can be improved by removing unnecessary labels and by adding an explanation of the asterisks on bars (P value?) and of the error bars (mean ± SD?) to the figure legends.

According to reviewer’s comment, we improved Figure 2 and 3,  and added P value in p.4,l.10-12 and mean ± SD in Table 2 and 3.

2. On Page 6, Line 147, it would be helpful if the authors could state precisely the percentage of reduction in elution after modifications as compared to control. The comment on "quantitative comparison" of test results applies here and elsewhere throughout the text.

According to reviewer’s comment, we added "quantitative comparison" of test results in Table 2 and 3.

Reviewer 2 Report

Going through the overall thesis report, the authors investigated suppression of the metal ion elution from orthodontic SS wire by formation of self-assembled monolayer. The theme is interesting and valuable. However, some concerns should be addressed fully in revising the manuscript. The authors might as well consider the following comments in the manuscript.

Major points

  1. Organization of paper should be reconsidered. It is recommended to divide result and discussion. Doing so would clearly indicate what the author is trying to assert.
  2. In overall report, it only says that the Cr and Ni metal ion elution can cause allergic reactions. Orthodontic SS wire is a medical device that has been used for years without major problems. It seems better to emphasize the specific human body reaction by that metal ion elution. More reference seems to be necessary and it is better to emphasize in introduction.
  3. Page 3 line 91 to 102: The correlation between surface wettability and metal ion elucidation is unclear. It is obvious that the long alkyl phosphate is more hydrophobic and turns hydrophilic when carboxyl group is added. Please explain the results of wettability. How can it be drawn out clinically meaningful?
  4. Page 4 Figure 2: Graph shows that there is small difference in (d) and (f), 0.1M lactic acid and saline respectively. It seems to be the experiment for paper’s sake. Does SAM coating really effectively can be applied in clinic? If the result is clinically important, it should be described and emphasized in discussion.
  5. Page 6 line 147: Author stated that the Cr and Ni ions were significantly decreased. Please describe statistical method to make that statement.
  6. Page 6 line 150: How the modification of SS wire can contribute to the antibacterial properties? Explain some reference about the statement specifically. This might helpful to readers.

Minor concerns

  1. Page 2 line 52 to 59: Author mentioned about four aqueous solutions, “HCl, phosphoric acid, lactic acid, and saline”. But the information about the process of preparing the solution and the manufacturer is omitted. It is recommended to give more information about pH of solutions and normal saliva. It will make this thesis report more clinically and convincing.
  2. Page 6 Figure 3: box of ‘plot area’ should be vanished.

Author Response

We are really sorry the delay of response. Because of the failure of the pH meter, we spent a delay for pH measurement of immersion solution.

According to reviewer’s comment, manuscript is checked by native English speaker (written by red letters).

Major points

1. Organization of paper should be reconsidered. It is recommended to divide result and discussion. Doing so would clearly indicate what the author is trying to assert.

According to reviewer’s comment, we divided results and discussion.

2. In overall report, it only says that the Cr and Ni metal ion elution can cause allergic reactions. Orthodontic SS wire is a medical device that has been used for years without major problems. It seems better to emphasize the specific human body reaction by that metal ion elution. More reference seems to be necessary and it is better to emphasize in introduction.

According to reviewer’s comment, we added description in p.6,l.16-27.

3. Page 3 line 91 to 102: The correlation between surface wettability and metal ion elucidation is unclear. It is obvious that the long alkyl phosphate is more hydrophobic and turns hydrophilic when carboxyl group is added. Please explain the results of wettability. How can it be drawn out clinically meaningful?

According to reviewer’s comment, We added the description in p.5,l.29-p.6,l.6. In this study, we considered that contact angle is one of the simple way to indicate SAM formation which the surface property was changed by.

4. Page 4 Figure 2: Graph shows that there is small difference in (d) and (f), 0.1M lactic acid and saline respectively. It seems to be the experiment for paper’s sake. Does SAM coating really effectively can be applied in clinic? If the result is clinically important, it should be described and emphasized in discussion.

According to reviewer’s comment, we added description in p.6,l.26-p.7,l.14

In this study, we varied the elution measurement as an accelerated simulation. For the purpose, the immersion condition was reduced gradually. As the immersion period expanded from a week to 4 weeks on 0.1 M lactic acid, the eluted ions were able to be detected. To indicate these phenomena, we showed condition (d) and (f).

5. Page 6 line 147: Author stated that the Cr and Ni ions were significantly decreased. Please describe statistical method to make that statement.

According to reviewer’s comment, we added description in p.4,l.10-12

6. Page 6 line 150: How the modification of SS wire can contribute to the antibacterial properties? Explain some reference about the statement specifically. This might helpful to readers.

According to reviewer’s comment, we added description in p.6,l.7-15.

Minor concerns

1. Page 2 line 52 to 59: Author mentioned about four aqueous solutions, “HCl, phosphoric acid, lactic acid, and saline”. But the information about the process of preparing the solution and the manufacturer is omitted. It is recommended to give more information about pH of solutions and normal saliva. It will make this thesis report more clinically and convincing.

In this study, HCl is used as a typical acidic solution. Phosphoric acid is used for edging treatment of enamel surface in clinical environment and lactic acid is generated in oral environment by metabolism. According to reviewer’s comment, we also added description in p.2,l.14-15.

2. Page 6 Figure 3: box of ‘plot area’ should be vanished.

According to reviewer’s comment, we improved Figure 3,

Reviewer 3 Report

This work is related with important scientific and clinical problem, however I feels to be obliged to point out the most important imperfections affecting his assessment.

  1. The methodology has been described too generally. The methodology of making samples should allow the experiment to be repeated, and this is not the case. The description of research methods lacks a lot of information, e.g. regarding the number of samples tested or the number of measurements on each of them.
  2. There is no statistical analysis of some results, statistical analyses were not mentioned in methodology section
  3. What were the criteria for selecting of the used conditioning solutions? The research was carried out in an acidic environment (concentrations should be listed in methodology section), and this should be linked with the properties prevailing in the mouth of the mouth. The authors should explain this.
  4. The is no discussion in this work (where are citations?), which is obligatory for scientific publications in recognized journals. However, there is only a brief presentation of the results without devoting to the causes and effects of the observed relations. It is disapointing.
  5. The figure 2 and 3 is completely chaotic and sloppy prepared.
  6. Why the ion emission for ODPA–SAM samples was teste, but for HOOC-DPA-SAM were was not analyzed? Why the ion emission after 4w was analyzed only for lactic acid solution?
  7. Why the condition of the surfaces of the samples were not analized after conditioning?
  8. Scheme 2 and 4 – it is your idea/investigation or it based on literature- knowledge?
  9. Why the conditions for he polyamine-modified dental wire were different than for other (anly one storing solution)?
  10. Some of the research methods that have been used to characterize the polymer layer are not mentioned in the methodology.
  11. Why did you study the contact angle? You don't write anything about it and you don't connect it with anything.
  12. The conclusion that one of the layers show the antibacterial property is not proven by you. Just because the shell material itself has such properties doesn't mean that your layer show it. This is only speculation.

In summary, the work is chaotic. The experiment is also inconsistent and does not allow full recognition of the subject of the study. In its present form it is not suitable for publication.

Author Response

We are really sorry the delay of response. Because of the failure of the pH meter, we spent a delay for pH measurement of immersion solution.

According to reviewer’s comment, manuscript is checked by native English speaker (written by red letters).

1.The methodology has been described too generally. The methodology of making samples should allow the experiment to be repeated, and this is not the case. The description of research methods lacks a lot of information, e.g. regarding the number of samples tested or the number of measurements on each of them.

According to reviewer’s comment, we added description in p.2,l.22-p.4,l.13.

2. There is no statistical analysis of some results, statistical analyses were not mentioned in methodology section.

According to reviewer’s comment, we added description in p.4,l.10-13.

3. What were the criteria for selecting of the used conditioning solutions? The research was carried out in an acidic environment (concentrations should be listed in methodology section), and this should be linked with the properties prevailing in the mouth of the mouth. The authors should explain this.

In this study, HCl is used as a typical acidic solution. Phosphoric acid is used for edging treatment of enamel surface in clinical environment and lactic acid is generated in oral environment by metabolism.

According to reviewer’s comment, we added description in p.2,l.13-14.

4. The is no discussion in this work (where are citations?), which is obligatory for scientific publications in recognized journals. However, there is only a brief presentation of the results without devoting to the causes and effects of the observed relations. It is disapointing.

According to reviewer’s comment, we divided result and discussion.

5. The figure 2 and 3 is completely chaotic and sloppy prepared.

When we submitted at first time, something happened on conversion process of these figures, unfortunately. In this time, we submit the original figures as a PDF file. According to reviewer’s comment, we improved Figure 2 and 3.

6. Why the ion emission for ODPA–SAM samples was teste, but for HOOC-DPA-SAM were was not analyzed? Why the ion emission after 4w was analyzed only for lactic acid solution?

According to reviewer’s comment, we added description in p.5,l.28-p.6,l.5. 

As HOOC-DPA-SAM is an intermediate product for polyamine-SAM, the ion elution was not analyzed in this study. And we carried out the elution measurement as an accelerated simulation in this study. In this viewpoint, the immersion condition was reduced gradually. 1 M lactic acid for a week and 0.1 M for 4 weeks are the weakest conditions. Thus, both of these conditions are used for polyamine-SAM.

7. Why the condition of the surfaces of the samples were not analized after conditioning?

After the immersion, the surface was analyzed by SEM and EDS. However, any clearly change were not observed. When the sample was immersed to 0.2 M HCl for a week, some pitting corrosion were observed as shown in Fig. S2. We added the description in p.7,l.2-8.

8. Scheme 2 and 4 – it is your idea/investigation or it based on literature- knowledge?

According to reviewer’s comment, we added reference 24 in p.5,l.29-30, and 25 in p.6,l.4-5 for Scheme 2 and 3, respectively.

9. Why the conditions for he polyamine-modified dental wire were different than for other (anly one storing solution)?

When the condition of an accelerated simulation of elution measurements were varied in this study, 1 M lactic acid for a week and 0.1 M for 4 weeks were the weakest conditions. Thus, polyamine-SAM samples were immersed under both conditions and the solution are analyzed.

10. Some of the research methods that have been used to characterize the polymer layer are not mentioned in the methodology.

According to reviewer’s comment, we added description in p.3,l.23-p.4,l.2.

11. Why did you study the contact angle? You don't write anything about it and you don't connect it with anything.

In this study, we considered that contact angle is one of the simple way to indicate SAM formation which the surface property was changed by. We added the description in p.5,l.29-p.6,l.6.

11. The conclusion that one of the layers show the antibacterial property is not proven by you. Just because the shell material itself has such properties doesn't mean that your layer show it. This is only speculation.

In this study, we were not able to carry out the antibacterial property of polyamine-SAM. However, some previous study indicated the antibacterial effect by amino group presentation on the surface as shown in ref. [26-30]. And we added description in p.6,l.7-15.

Round 2

Reviewer 2 Report

The authors made the progress.

Author Response

According to reviewer’s comment, we added description on introduction (p.1,l.30-p.2,l.9, and p.2,l.20-23.)

Reviewer 3 Report

The authors made significant corrections to the text or provided explanations.

I suggest, however, that you can think again about your conclusions. The reviewer understands very good that the issue of bacterial properties is based on literature data, but in this case it is not a conclusion from your work. If you necessarily want to set it (although in my opinion it does not contribute much on this work), please clearly state that this is indicated by the results of other published studies, not by your investigations, becouse it is confusing for readers. The same should be done in the "abstract" .

Author Response

The authors made significant corrections to the text or provided explanations.

I suggest, however, that you can think again about your conclusions. The reviewer understands very good that the issue of bacterial properties is based on literature data, but in this case it is not a conclusion from your work. If you necessarily want to set it (although in my opinion it does not contribute much on this work), please clearly state that this is indicated by the results of other published studies, not by your investigations, becouse it is confusing for readers. The same should be done in the "abstract"

According to reviewer’s comment, we removed description for polyamine-SAM on abstract and conclusions (p.1,l.12-20., and p.7,l.16-22.)
